# Rapid bacterial evaluation beyond the colony forming unit in osteomyelitis

Qi Sun[1], Kimberley Huynh[1], Dzenita Muratovic[1], Nicholas J Gunn[1], Anja R Zelmer[1], Lucian Bogdan Solomon[1,2], Gerald J Atkins[1]*[†], Dongqing Yang[1]*[†]

[1]Centre for Orthopaedic & Trauma Research, Faculty of Health and Medical Sciences, University of Adelaide, Adelaide, Australia; [2]Department of Orthopaedics and Trauma, Royal Adelaide Hospital, Adelaide, Australia

**Abstract** Examination of bacteria/host cell interactions is important for understanding the aetiology of many infectious diseases. The colony forming unit (CFU) has been the standard for quantifying bacterial burden for the past century, however, this suffers from low sensitivity and is dependent on bacterial culturability in vitro. Our data demonstrate the discrepancy between the CFU and bacterial genome copy number in an osteomyelitis-relevant co-culture system and we confirm diagnosis and quantify bacterial load in clinical bone specimens. This study provides an improved workflow for the quantification of bacterial burden in such cases.

*For correspondence:
gerald.atkins@adelaide.edu.au (GJA);
dongqing.yang@adelaide.edu.au (DY)

[†]Co-senior authors

Competing interest: The authors declare that no competing interests exist.

## eLife assessment

This **fundamental** study addresses discrepancies in determining bacterial burden in osteomyelitis as determined by culture and enumeration using DNA. The authors present **compelling** data demonstrating the emergence of discrepancies between CFU counts and genome copy numbers detected by PCR in *Staphylococcus aureus* strains infecting osteocyte-like cells. The observations represent a substantial addition to the field of musculoskeletal infection, with possible broad applicability and clinical benefit to other infectious diseases.

## Introduction

Co-culturing mammalian host cells with bacteria in vitro is an important tool for examining the host–pathogen interaction in infectious disease modelling (*Hofstee et al., 2020*). One of the essential readouts is the intracellular bacterial burden carried by the infected host cells. The most commonly used method for enumerating such bacterial load is counting the colony forming unit (CFU) number by agar plate culturing following the lysis of host cells and release of bacteria (*Staley and Konopka, 1985*). However, the reliability of such measurements has long been recognised to be impacted by various factors, including, but not limited to, host cell/tissue type, bacterial strain, bacterial load, and culturing conditions for CFU enumeration (*Chang et al., 1995*; *Davis et al., 2005*). Osteomyelitis (OM), an infectious disease with pathogen-mediated infection of bone tissues, features a considerable proportion of infected but culture-negative cases, ranging between 20% and 40% (*Bejon et al., 2010*; *Yoon et al., 2017*; *Kim et al., 2015*). This clear deficiency in CFU evaluation creates difficulties for the accurate diagnosis and therefore treatment of OM.

## Results and discussion

*Staphylococci*, including coagulase-negative species, such as *S. epidermidis* and *S. aureus*, comprise the major pathogens in adult OM (*Kavanagh et al., 2018*). In this study, we employed a previously

established osteocyte-like cell model, differentiated SaOS2 (SaOS2-OY) and the single most common causative pathogen in OM, *S. aureus*, to simulate the in vitro infection of osteocytes, the most abundant cell type in bone tissue (*Prideaux et al., 2014*; *Gunn et al., 2021*). Two previously characterised *S. aureus* strains, a high virulence (methicillin-resistant *S. aureus*; MRSA) strain, WCH-SK2 (SK2), shown previously to establish an intracellular infection in human osteocytes (*Yang et al., 2018*) and a low virulence (methicillin-sensitive *S. aureus*; MSSA) strain, WCH-SK3 (SK3) (*Bui and Kidd, 2015*), were chosen for experimentation. In addition to CFU enumeration to quantify bacterial number, a polymerase chain reaction (PCR) approach measuring genomic DNA copies represented by a single-copy gene within the *S. aureus* genome, was also performed in parallel. In the past, real-time quantitative PCR (qPCR) assays have been widely used for the quantification of bacteria. However, the working principle of standard qPCR is dependent on the establishment of a standard curve using a known number of bacterial gene copies and the fitting of test samples into a linear regression to estimate the bacterial number for each assay (*Kralik and Ricchi, 2017*; *Smith and Osborn, 2009*). Here, we utilised digital droplet PCR (ddPCR) for this measurement. By its nature, the latter technique offers the advantage of counting the absolute target sequence copy number within the sample and removes the need for normalisation (*Hindson et al., 2013*). For all PCR-based assays, the quality of DNA containing the target sequence for amplification and quantification is critical for the sensitivity and reproducibility of readout. Therefore, methods for DNA preparation from both human and bacterial cells were examined. Here, we introduced the usage of DirectPCR Lysis Reagent (Direct buffer), a lysis buffer that maximises the release of genomic DNA from samples and is compatible with the downstream PCR analysis from cell culture lysate without any purification step. With the application of Direct buffer for DNA preparations in comparison to a commonly used DNA extraction spin column kit (DNA kit), the absolute bacterial genome copy number quantified by ddPCR was measured to be 5-fold higher in SaOS2-OY samples and 100-fold higher in samples from two strains of *S. aureus*, SK2 and SK3; better overall reproducibility among biological replicates was also achieved using Direct buffer over the standard approach (*Figure 1A–C*). Serial 10-fold dilutions of bacterial log-phase suspension cultures were used to generate DNA lysates for testing the extent of DNA release. Plotting genome copy number against CFU from the same cultures showed near perfect linear regression relationships (*Figure 1D, E*), confirming the complete and highly reproducible release of bacterial DNA. With this validating readout, we sought to determine if the elimination of purification steps by using the Direct buffer might minimise the sample loss associated with the utilisation of conventional column-based extraction methods (*Schmitz et al., 2021*). Furthermore, the release of genomic material by this reagent appeared to be complete and unbiased in our experiments, which led to higher consistency of inter- and intra-assays.

Using the combined approach of direct lysis DNA preparation and ddPCR quantification, we then examined *S. aureus* persistence in the intracellular environment in infected SaOS2-OY cells. Following infection, the SaOS2-OY and *S. aureus* co-cultures were lysed to break down host cell structures for the release of intracellular *S. aureus* for CFU development on agar plates. Consistent with previous findings, significant reductions ranging between $10^3$- and $10^6$-fold in cultured CFU on agar plates over the 5-day co-culture time period were observed in both SK2- and SK3-infected SaOS2-OY groups (*Figure 2A–F*). This trend was also found with low multiplicities of infection (MOI) (~$10^7$ total CFU/well, *Figure 2C, E*) and high MOI (~$10^8$ total CFU/well, *Figure 2D, F*) for initial bacteria-inoculated groups with either strain. However, when total DNA preparations from the whole culture lysates were taken for ddPCR quantification of bacterial genome copy number, differences in the dynamic changes of bacterial load were found. In the high virulence strain SK2-infected group, the genome copy numbers by 5-day post-infection were maintained at the levels of $10^7$ and $10^8$ cells/well for the low and high MOI groups, respectively (*Figure 2C, D*), similar to the original input levels.

However, the low virulence strain SK3-infected group demonstrated a decrease in genome copy counts. Interestingly, regardless of the initial input amount, by 5-days post-infection, the remaining SK3 genome counts dropped to approximately $10^6$ cells/well for both groups (*Figure 2E, F*). The quantification of human genome copy number was also performed in all groups to estimate the remaining host cells in culture and this also served as an indicator of host cell viability. With the low MOI co-cultures, the human genome copy counts in the presence of either strain of *S. aureus* were not statistically different from uninfected controls, across 1- and 5-days post-infection. When the initial bacterial inoculum was increased 10-fold to $10^8$ cells/well, reductions in the human genome copies

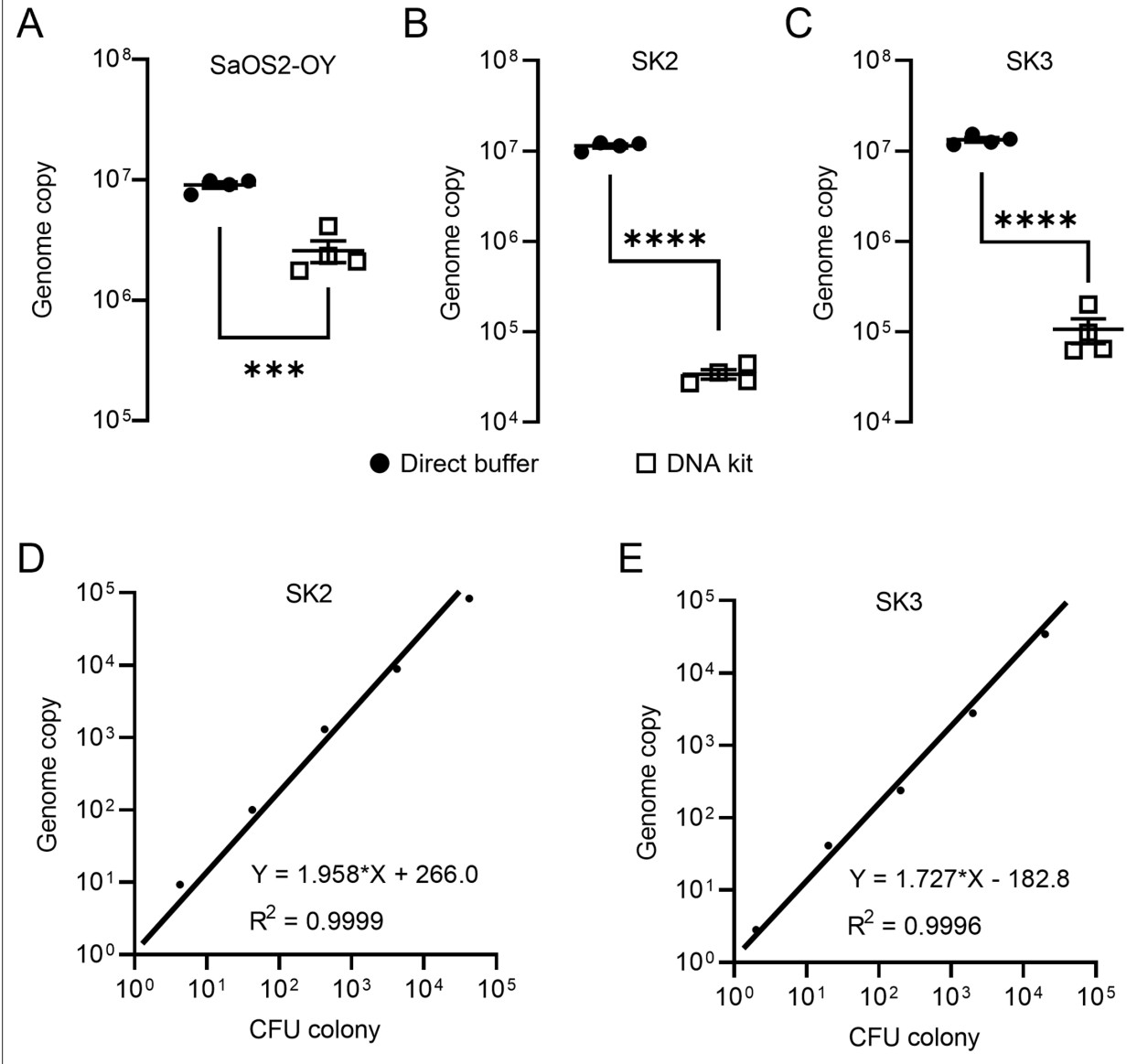

**Figure 1.** Validation of DNA preparations from SaOS2-OY cells and *S. aureus*. Digital droplet PCR (ddPCR) genome counting of SaOS2-OY (**A**), SK2 (**B**), and SK3 (**C**), comparing the Direct buffer approach (●) and a standard DNA kit (□) (four biological replicates with mean and standard errors were shown on graphs, ***p < 0.001 and ****p < 0.0001); demonstration of complete bacterial chromatin release from SK2 (**D**) and SK3 (**E**): correlation with CFU plating. Each data point represented the comparison of one colony-forming unit (CFU) recovery and one total DNA measurement; three independent experiments were carried out for each strain and similar results were achieved; representative results of one experiment are presented.

of ~30% and 50% were measured in 1- and 5-day post-infection groups, respectively; such a trend was found to be similar in both SK2- and SK3-infected groups. Overall, our observations demonstrated profound differences in the readout of the presence of intracellular bacteria when using CFU and ddPCR quantification methods. Such inconsistency between these readouts in our model was as high as $10^6$-fold, with interaction of factors including initial bacterial load and strain variation. In contrast, comparison of the same quantification methods performed on bacterial suspension cultures yielded differences under twofold, indicated by the respective slope factor (less than 2) in the linear regression curves for both SK2 and SK3 strains (see *Figure 1D, E*). Together, our results indicate the dramatic change in bacterial culturability when comparing growth in ideal microbial suspension culture conditions and the growth-limiting intracellular environment. Therefore, we propose that the interaction between the host cell response to infection and pathogen adaptation to the intracellular environment will compromise the reliability of the standard CFU counting method alone for the evaluation of

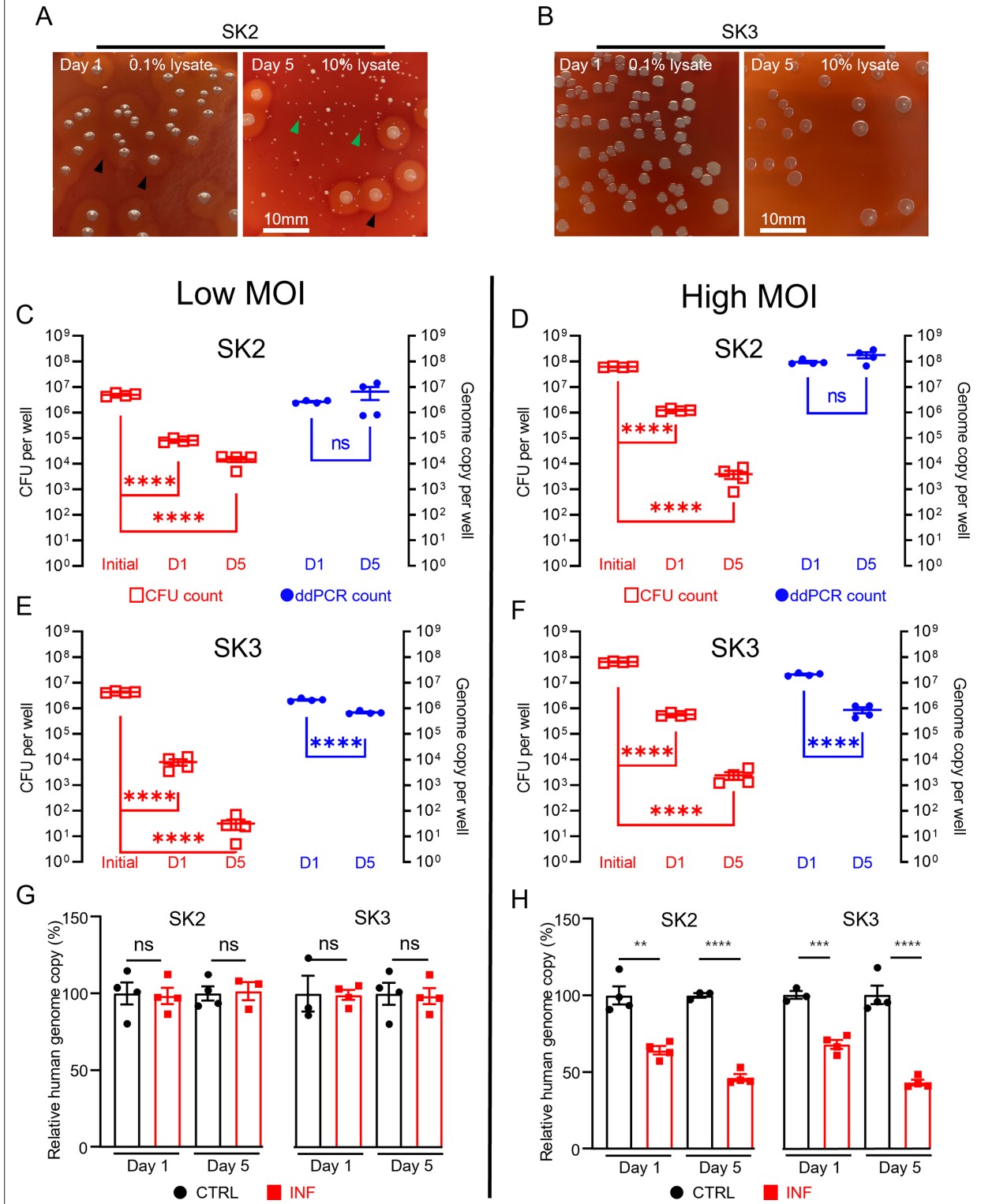

**Figure 2.** Measurement of intracellular *S. aureus* in SaOS2-OY cells in vitro. Colony-forming unit (CFU) recovery of SK2 (**A**) and SK3 (**B**) from host SaOS2-OY cells, with haemolysis reactions (◀) and numerous small colony variants (◀) shown in SK2 group; quantification of SK2 and SK3 using CFU count (☐, left Y) and digital droplet PCR (ddPCR) count (●, right Y) from low (**C, E**) and high (**D, F**) multiplicities of infection (MOI) groups; relative human genome copy (%) with CTRL (●) vs. INF (■) quantified by ddPCR from low (**G**) and high (**H**) MOI groups (four biological replicates with mean and standard errors are shown; **p < 0.01, ***p < 0.001, ****p < 0.0001; ns = not significant).

bacterial persistence in a bacteria/host cell co-culture experiment. However, phenotypic variations were also observed in the recovered colonies from agar plating, with variations in haemolytic activity and small colony variant (SCV) formation, particularly in the SK2 group (*Figure 2A*). SCV variants of *S. aureus* feature reduced metabolic activity and growth but are an indication of adaptation and the establishment of a chronic infection. This is consistent with the findings of reductions in CFU recovery but maintained levels of detectable genome copies, at least in the SK2-infected group.

To examine the applicability of this workflow to a clinical setting, we next extended the tests to clinical human bone specimens. Three clinically culture-negative periprosthetic joint infection (PJI) cases were chosen for analysis. All three cases were confirmed infected by clinical observation for typical PJI symptoms by clinicians, using the current gold-standard Musculoskeletal Infection Society (MSIS) criteria (*Parvizi et al., 2018*); while each patient presented with a sinus tract communicating with their prosthesis, a major diagnostic criterion, neither the culturing of the synovial membrane biopsies nor synovial fluid around the prosthesis returned any positive cultures from a clinical laboratory. The bone tissue sections from the three patients were examined by Masson's trichrome staining in comparison to tissue section from an osteoarthritis (OA) patient as a non-infection control, to examine bone matrix degradation, according to our previous study (*Ormsby et al., 2021*). With this method, the red colour indicates intact bone matrix collagen and the blue colour corresponds to degraded collagen. Within all the histological observations, matrix degradation was found in sections of all three PJI patients (*Figure 3B–D*), whereas the staining of the control OA patient bone indicated intact collagen (*Figure 3A*), consistent with infection in the three PJI patient specimens (*Ormsby et al., 2021*). In addition to the gold-standard diagnostic approach and histological staining, we performed CFU analysis using homogenised bone tissue, and negative culture results were also recorded. In this study, the samples containing total genomic DNA were prepared from histological sections in conjunction with the use of the DirectPCR Lysis reagent. The prepared DNA samples were subjected to PCR amplification using a primer set targeting a highly conserved bacterial genomic region, elongation factor Tu (*tuf*) (*Li et al., 2012*), for the enrichment of bacterial signal within host/pathogen genomic mixture. The generated amplicons were then sequenced using an Oxford Nanopore Technology (ONT) MinION sequencer for the identification of bacterial species. All three patient bone DNA samples were confirmed to be bacterial genome-positive, with combinations identified of the coagulase-negative staphylococcal species *S. haemolyticus*, *S. hominis*, and *S. epidermidis* (*Figure 3E–G*). The respective total bacterial load for each of the samples was then quantified by ddPCR. Each bone sample examined contained between $2 \times 10^4$ and $1 \times 10^6$ bacterial genomic copies per million human genomic copies (*Figure 3E–G*). To confirm that the detected bacteria species were not introduced during the handling procedures, bone specimens from five primary total hip replacement (non-infected) patients were processed at the same time along with the PJI specimens; PCR reactions with negative *tuf* amplification for these bone samples indicated that operational contamination was unlikely (*Figure 3—figure supplement 1*).

In this study, by the analysis of genomic material, we have addressed the ineffectiveness of the classical bacterial CFU development approach in determining and quantifying a confirmed infection by both an in vitro experimental model and by examining human clinical cases. With the introduction of a direct DNA release approach, less handling is required while delivering increased accuracy. The addition of ddPCR provided the advantage of achieving absolute quantification without the need for a PCR standard curve, which is both cost and time effective. For the purposes of unknown pathogen diagnosis in clinical cases, the exact bacterial species readout is required from sequencing the generated amplicons. Here, the utilisation of ONT dramatically lowered the equipment and skill demands for performing this analysis, since the MinION sequencer is a portable device and the sequencing results are analysed by the on-board software with little bioinformatics requirement. Noted here, the usual whole-genome sequencing approach might not be suitable for this application, as in many of the clinical specimens, the amount of human genome can be orders of magnitude higher than the level of bacterial genomic material. Sequencing and analysing the relatively rare bacterial DNA present in this scenario is considerably more lengthy and costly. Therefore, a PCR approach was chosen to amplify the bacterial signal for detection. In this study, the bacterial gene *tuf* was chosen as the target sequence to generate PCR amplicons instead of the commonly used *16S* rRNA gene sequence because of its greater discriminatory power (*Hwang et al., 2011*). Here, we took *tuf* sequences from *S. aureus* and the three coagulase-negative *staphylococcal* species identified in the clinical specimens

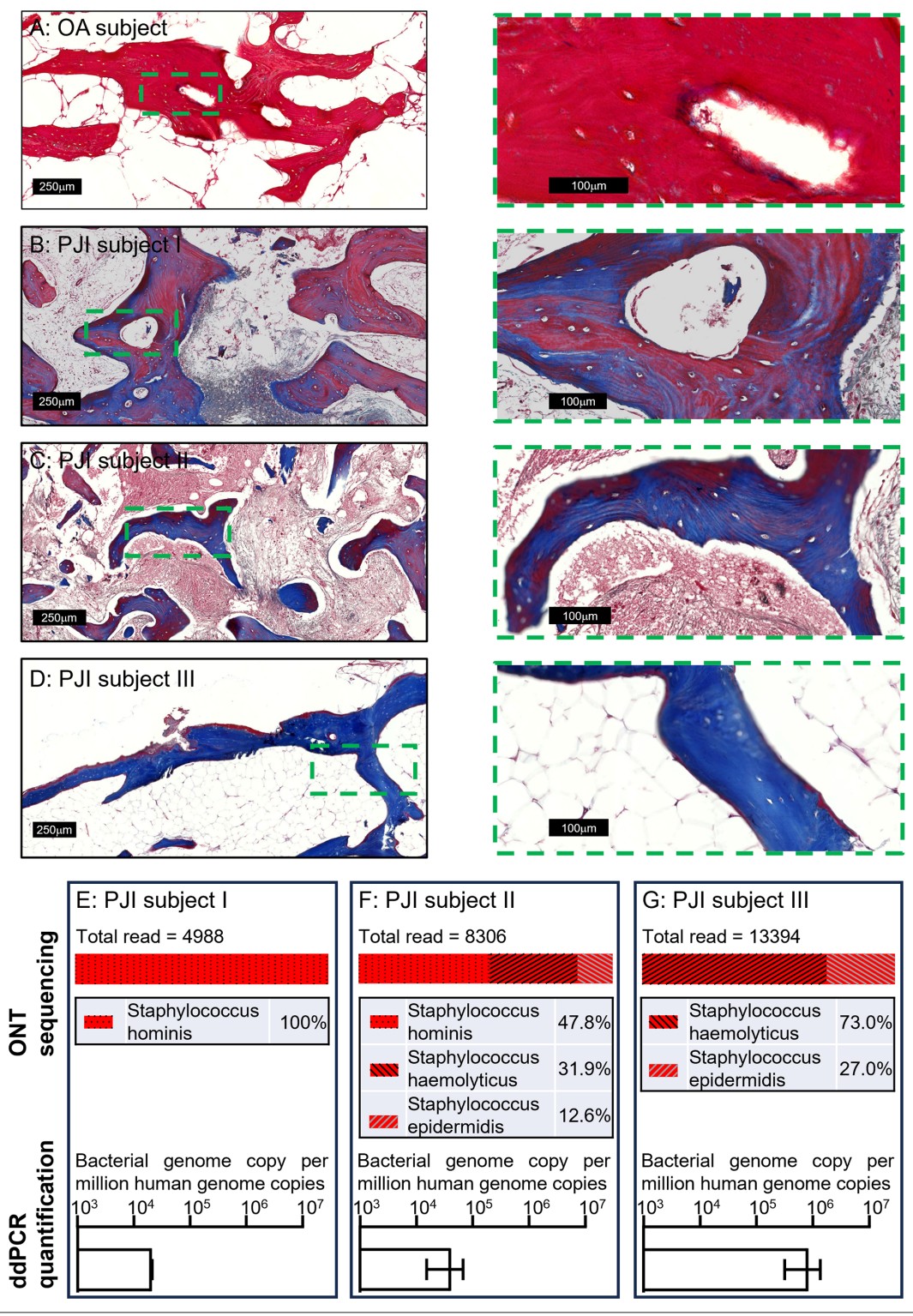

**Figure 3.** Pathogen diagnosis in clinical PJI bone specimens. Masson's trichrome staining of bone tissue sections of an osteoarthritis (OA) subject (**A**) and culture-negative periprosthetic joint infection (PJI) subjects I–III (**B–D**). Pathogen profiling using total DNA of bone specimens from the above three PJI patients (**E–G**) by ONT sequencing for the readout of pathogen species and digital droplet PCR (ddPCR) to quantify bacterial load as a ratio of bacterial: human genomic copies (error bars shown are the combination of the standard error of the mean

*Figure 3 continued*

from two individual DNA preparations and the device generated error from each of the ddPCR runs, with over 15,000 droplets read per sample).

The online version of this article includes the following figure supplement(s) for figure 3:

**Figure supplement 1.** Polymerase chain reaction (PCR) analysis of bone samples from primary total hip replacement cases.

**Figure supplement 2.** Multi-sequence alignment among *S. aureus*, *S. epidermidis*, *S. haemolyticus*, and *S. hominis*.

---

for cross-comparisons (*Figure 3—figure supplement 2A*). Within the designated amplified sequence of only 271 bp, 11–22 (or 4.1–8.1% differences) mismatches were identified between these four species (*Figure 3—figure supplement 2B*). For achieving equivalent differences in *16*S sequences among the four strains, PCR amplicons with over 1000 bp in length are required (data not shown). Again, the strategy of analysing a short amplicon not only offers reduced reaction time, more importantly, it minimises the chance of generating non-specific products by shortening the elongation step. The listed primer set targeting the *tuf* gene was confirmed to cover most of the species in genus *Staphylococcus in silico*. In practice, it is challenging to find universal primers that cover the entire bacterial domain. Hence, for future real-world tests in clinical diagnosis, the strategy of using multiple primer sets to cover common PJI pathogens might need to be considered. Together, this workflow is potentially applicable as a point-of-care diagnostic method with the prospect of a rapid turnover: within hours compared to the current procedure in the magnitude of days.

To conclude, for the purpose of quantifying the intracellular bacterial load in a host–pathogen co-culture setting, we have developed a workflow with the advantages of: (1) being rapid and labour-saving, with the elimination all DNA extraction steps by virtue of using the direct lysis approach; (2) minimisation of potential sample loss encountered with conventional DNA isolation methods, enabling better accuracy and reproducibility; (3) absolute genome copy number quantification by ddPCR for achieving superior consistency of intra-assay experiments; absolute bacterial load readout is particularly useful to track infection status with longitudinal serial samples from PJI patients; (4) sequencing analysis using the ONT MinION platform has the potential to become a point-of-care diagnostic approach when performing unknown pathogen identification in a clinical setting. Nevertheless, we suggest that the CFU plating method should not be omitted as it allows the evaluation of bacterial phenotypic adaptation in such experimentation.

## Materials and methods
### SaOS2 cell culture and differentiation
The human osteosarcoma cell line SaOS2 differentiated to an osteocyte-like stage was employed for performing bacteria/host co-cultures, as previously described (*Gunn et al., 2021*). Cells were seeded at a density of $2 \times 10^4$ cells/well in 48-well plates and cultured at 37°C/5% $CO_2$, in growth media consisting of alpha modification of minimum essential medium (αMEM) (Thermo Fisher, Victoria, Australia) supplemented with 10% vol/vol foetal calf serum (FCS) (Thermo Fisher), 2 mM L-glutamine (Thermo Fisher), and 1 U/ml penicillin/streptomycin (Thermo Fisher). At 90% confluence, SaOS2 cultures were switched to osteogenic differentiation medium comprising αMEM supplemented with 5% vol/vol FCS, 100 mM ascorbate 2-phosphate (Sigma-Aldrich, St Louis, USA), 1.8 mM potassium di-hydrogen phosphate (Sigma-Aldrich), 1 mM 4-(2-hydroxyethyl)-1-piperazineethanesulfonic acid (HEPES) (Thermo Fisher), 2 mM L-glutamine, and 1 U/ml penicillin/streptomycin. Cells were then maintained under differentiation conditions for 28 days to achieve an osteocyte-like phenotype (SaOS2-OY) (*Prideaux et al., 2014*).

### *S. aureus* preparation
Two pre-characterised *S. aureus* strains, an MRSA strain, WCH-SK2 (SK2) (*Bui and Kidd, 2015*) and an MSSA strain, WCH-SK3 (SK3), were grown in Terrific Broth (Thermo Fisher) on a 37°C/200 rpm rocking platform to achieve log-phase suspension cultures individually. Bacteria were pelleted by 3000 × *g* for 5 min centrifugation and then resuspended in sterile phosphate-buffered saline (PBS) to estimate

cell number by the optical density at 600 nm light absorption. The two bacteria suspensions were then taken to undergo 10-fold serial dilutions and the diluents were individually plated on blood agar containing 10% vol/vol defibrinated sheep blood (Thermo Fisher) to determine the CFU number from the original bacterial suspension preparations. The calculated CFU numbers from agar plates were compared to the ddPCR-quantified genome copy numbers.

## Host cell infection by co-culturing of SaOS2-OY and *S. aureus*

For host cell infection, bacterial suspension cultures were resuspended in sterile PBS, as described above, to achieve the density for low and high MOI. The differentiation media for SaOS2-OY was removed and the cells washed twice with PBS. The resulting bacterial inoculums were added to SaOS2-OY cells to allow invasion for 1 hr at 37°C/5% $CO_2$. Post-infection, cultures were washed twice with PBS and incubated at 37°C/5% $CO_2$ with 20 µg/ml lysostaphin (Sigma-Aldrich) in antibiotic-free media for 24 hr to eliminate extracellular bacteria before replenishment with fresh differentiation medium. Supernatants were streaked on agar plates to verify the absence of extracellular bacteria at 24- and 120-hr post-infection.

## Measurements of intracellular bacterial number by CFU counting

The intracellular bacteria numbers were determined at 24- and 120-hr post-infection time points by spreading serial dilutions of cell lysates on blood agar and culturing at 37°C/5% $CO_2$ for 48 hr.

## DNA extraction from host cells and bacteria

For genomic quantification, total DNA was isolated using either DNeasy Blood & Tissue Kits (Qiagen Inc, VIC, Australia) or DirectPCR Lysis Reagent (Direct buffer) (Viagen Biotech Inc, CA, USA), as per the manufacturer's instructions. For the new approach of using the Direct buffer, 500 µl buffer together with 200 µg/ml proteinase K (Thermo Fisher) was used for the lysis of one SaOS2-OY sample in one well from a 48-well tissue culture plate or one pelleted bacterial sample prepared from serial dilutions of a suspension culture. Individual cell lysates were transferred to 1.5 ml reaction tubes, digested at 55°C for 35 min and then heat-inactivated at 85°C for 15 min to terminate enzymic digestion. The processed lysates were then ready for PCR analyses.

## Digital droplet polymerase chain reaction

The ddPCR assay was performed using QX200 Digital Droplet PCR System (Bio-Rad Laboratories, USA), as per the manufacturer's instructions. Primer sets targeting a human genome-specific sequence within the single-copy type X collagen (*COL10A1*) gene and a *S. aureus* genome-specific sequence within the sigma factor B (*sigB*) gene were used. The sequences of primer sets were:

> *COL10A1*: forward 5'-ccaccaggtcaagcagtcat-3', reverse 5'-gttggcactaacaagaggggt-3'
> *sigB*: forward 5'-ggggcaacaagatgaccatt-3', reverse 5'-tgccgttctctgaagtcgtg-3'

## Analysis of human bone tissues

All human studies received institutional research ethics approval (Royal Adelaide Hospital Human Research Ethics Committee Approval No. 14446). Bone biopsies were collected from patients undergoing either primary total hip replacement or revision total hip replacement surgery for PJI, with informed, written patient consent. Each bone specimen was separated into two parts. One of these was homogenised in a bead-beating tissue homogeniser (Bead Ruptor Elite, Omni International, Kennesaw, GA, USA), with two cycles of beating at 3 m/s for 30 s to disassociate the bone structure for the release of potentially viable bacteria. Bone homogenates were then sent to a clinical pathology laboratory (SA Pathology, Adelaide, Australia) for CFU analysis. Clinical soft tissue and fluid samples were sent to SA Pathology directly from surgery for pathogen analysis. The second piece of bone biopsy was processed and de-mineralised using OSTEOSOFT (Sigma-Aldrich) solution for standard paraffin embedding procedure. Post-embedding, the bone tissues were sectioned under DNase/RNase-free conditions. One 5 µm section from each bone specimen was used for Masson's Trichrome staining, as described in our previous study (*Ormsby et al., 2021*), and six sections were pooled together in DirectPCR Lysis Reagent for the isolation of total genomic DNA, as described above. Prepared DNA samples were then subjected to (1) conventional PCR amplification for an ONT

sequencing readout (MinION sequencer, Oxford Nanopore Technology, Oxford, UK), according to the manufacturer's instructions; (2) ddPCR for the absolute quantification for both human and bacterial genome copy number. The primer sets targeting the human *COL10A1* genomic sequence, as listed above, and bacterial *tuf* sequence primers (forward 5'-ttctcaatcactggtcgtgg-3', reverse 5'-ggag tatgacgtccaccttc-3') were used for measuring human and bacterial genome copies, respectively. The output sequencing data were analysed by Epi2ME software within the WIMP workflow (Oxford Nanopore Technologies) for bacterial taxonomical analysis.

## Statistics

The results were graphed as means ± standard errors of the means. The significance between two treatment groups was evaluated using two-tailed *T*-test using GraphPad Prism 10.2 (GraphPad Software, MA, USA), where applicable.

## Acknowledgements

This work was supported by a National Health and Medical Research Council of Australia (NHMRC) Ideas Grant scheme (ID 2011042) awarded to GJA and DY and an Australian Orthopaedic Association Research Grant awarded to GJA and LBS. Oxford Nanopore sequencing was supported by a University of Adelaide Faculty of Health and Medical Sciences infrastructure grant with technical assistance provided by Ms Thessa Kroes and Dr. Mark Corbett. QS was supported by a University of Adelaide Faculty of Health and Medical Sciences Postgraduate Research Scholarship.

## Additional information

### Funding

| Funder | Grant reference number | Author |
| --- | --- | --- |
| National Health and Medical Research Council | 2011042 | Gerald J Atkins<br>Dongqing Yang |
| Australian Orthopaedic Association | | Lucian Bogdan Solomon<br>Gerald J Atkins |
| University of Adelaide | Postgraduate Research Scholarship | Qi Sun |

The funders had no role in study design, data collection, and interpretation, or the decision to submit the work for publication.

### Author contributions

Qi Sun, Data curation, Formal analysis, Investigation, Methodology, Writing – original draft, Writing – review and editing; Kimberley Huynh, Data curation, Formal analysis, Investigation; Dzenita Muratovic, Resources, Data curation, Formal analysis, Validation, Investigation, Methodology, Writing – review and editing; Nicholas J Gunn, Validation, Investigation, Methodology; Anja R Zelmer, Validation, Investigation; Lucian Bogdan Solomon, Resources, Investigation, Methodology, Writing – review and editing; Gerald J Atkins, Conceptualization, Resources, Supervision, Funding acquisition, Project administration, Writing – review and editing; Dongqing Yang, Conceptualization, Resources, Formal analysis, Supervision, Funding acquisition, Validation, Investigation, Visualization, Methodology, Writing – original draft, Project administration, Writing – review and editing

### Author ORCIDs

Qi Sun ◆ http://orcid.org/0000-0003-4826-6724
Gerald J Atkins ◆ http://orcid.org/0000-0002-3123-9861
Dongqing Yang ◆ https://orcid.org/0000-0002-5471-6288

### Ethics

All human studies received institutional research ethics approval (Royal Adelaide Hospital Human Research Ethics Committee Approval No. 14446). Bone biopsies were collected from patients

undergoing either primary total hip replacement or revision total hip replacement surgery for PJI, with informed, written patient consent.

Reviewer #1 (Public Review): https://doi.org/10.7554/eLife.93698.3.sa1
Reviewer #2 (Public Review): https://doi.org/10.7554/eLife.93698.3.sa2
Author response https://doi.org/10.7554/eLife.93698.3.sa3

---

# Additional files

## Supplementary files
• MDAR checklist

## Data availability
DNA sequencing data was uploaded to Dryad (https://doi.org/10.5061/dryad.s4mw6m9f5); all other data generated or analysed during this study are included in the manuscript and supporting files.

The following dataset was generated:

| Author(s) | Year | Dataset title | Dataset URL | Database and Identifier |
|---|---|---|---|---|
| Yang D, Sun Q, Huynh K, Muratovic D, Gunn NJ, Zelmer AR, Solomon LB, Atkins GJ | 2024 | Data from: Beyond the colony-forming-unit: Rapid bacterial evaluation in Osteomyelitis | https://doi.org/10.5061/dryad.s4mw6m9f5 | Dryad Digital Repository, 10.5061/dryad.s4mw6m9f5 |

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
