## [Editor Report · eLife assessment]

This **fundamental** study addresses discrepancies in determining bacterial burden in osteomyelitis as determined by culture and enumeration using DNA. The authors present **compelling** data demonstrating the emergence of discrepancies between CFU counts and genome copy numbers detected by PCR in *Staphylococcus aureus* strains infecting osteocyte-like cells. The observations represent a substantial addition to the field of musculoskeletal infection, with possible broad applicability and clinical benefit to other infectious diseases.

---

## [Referee Report · Reviewer #1 (Public Review)]

Summary:

This work shows, based on basic laboratory investigations of in vitro grown bacteria as well as human bone samples, that conventional bacterial culture can substantially underrepresent the quantity of bacteria in infected tissues. This has often been mentioned in the literature, however, relatively limited data has been provided to date. This manuscript compares culture to a digital droplet PCR approach, which consistently showed greater levels of bacteria across the experiments (and for two different strains).

Strengths:

Consistency of findings across in vitro experiments and clinical biopsies. There are real-world clinical implications for the findings of this study.

Weaknesses:

No major weaknesses. Only 3 human samples were analyzed, although the results are compelling.

---

## [Referee Report · Reviewer #2 (Public Review)]

In this study, the authors address discrepancies in determining the local bacterial burden in osteomyelitis between that determined by culture and enumeration by DNA-directed assay. Discrepancies between culture and other means of bacterial enumeration are long established and highlighted by Staley and Konopka's classic, "The great plate count anomaly" (1985). Here, the authors first present data demonstrating the emergence of discrepancies between CFU counts and genome copy numbers detected by PCR in *S. aureus* strains infecting osteocyte-like cells. They go on to demonstrate PCR evidence that *S. aureus* can be detected in bone samples from sites meeting a widely accepted clinico-pathological definition of osteomyelitis. They conclude their approach offers advantages in quantifying intracellular bacterial load in their in vitro "co-culture" system.

WEAKNESSES

(A) My main concern here is the significance of these results outside the model osteocyte system used by this group. Although they carefully avoid over-interpreting their results, there is a strong undercurrent suggesting their approach could enhance aetiologic diagnosis in osteomyelitis and that enumeration of the infecting pathogen might have clinical value. In the first place molecular diagnostics such as 16S rDNA-directed PCR are well established in identifying pathogens that don't grow. Secondly, it is hard to see how enumeration could have value beyond in vitro and animal model studies since serial samples will rarely be available from clinical cases.

(B) I have further concerns regarding interpretation of the combined bacterial and host cell-directed PCRs against the CFU results. Significance is attached to the relatively sustained genome counts against CFU declines. On the one hand it must be clearly recognised that detection of bacterial genomes does not equate to viable bacterial cells with potential for further replication or production of pathogenic factors. Of equal importance is the potential contribution of extracellular DNA from lysed bacteria and host cells to these results. The authors must clarify what steps, if any, they have taken to eliminate such contributions for both bacteria and host cells. Even the treatment with lysotaphin may have coated their osteocyte cultures with bacterial DNA, contributing downstream to the ddPCR results presented.

STRENGTHS

(C) On the positive side, the authors provide clear evidence for the value of the direct buffer extraction system they used as well as confirming the utility of ddPCR for quantification. In addition, the successful application of MinION technology to sequence the EF-Tu amplicons from clinical samples is of interest.

(D) Moreover, the phenomenology of the infection studies indicating greater DNA than CFU persistence and differences between the strains and the different MOI inoculations are interesting and well-described, although I have concerns regarding interpretation.

---

## [Author Response]

The following is the authors’ response to the original reviews.

**Reviewer #1 (Public Review):**
Summary:This work shows, based on basic laboratory investigations of invitro-grown bacteria as well as human bone samples, that conventional bacterial culture can substantially underrepresent the quantity of bacteria in infected tissues. This has often been mentioned in the literature, however, relatively limited data has been provided to date. This manuscript compares culture to a digital droplet PCR approach, which consistently showed greater levels of bacteria across the experiments (and for two different strains).Strengths:Consistency of findings across in vitro experiments and clinical biopsies. There are real-world clinical implications for the findings of this study.Weaknesses:No major weaknesses. Only three human samples were analyzed, although the results are compelling.

We only put in three examples of clinical diagnosis to showcase the application of this method particularly to osteomyelitis. For further validation, larger cohort studies are required, which are currently underway.

**Reviewer #2 (Public Review):**
In this study, the authors address discrepancies in determining the local bacterial burden in osteomyelitis between that determined by culture and enumeration by DNA-directed assay. Discrepancies between culture and other means of bacterial enumeration are long established and highlighted by Staley and Konopka's classic, "The great plate count anomaly" (1985). Here, the authors first present data demonstrating the emergence of discrepancies between CFU counts and genome copy numbers detected by PCR in *S. aureus* strains infecting osteocyte-like cells. They go on to demonstrate PCR evidence that *S. aureus* can be detected in bone samples from sites meeting a widely accepted clinicopathological definition of osteomyelitis. They conclude their approach offers advantages in quantifying intracellular bacterial load in their in vitro "co-culture" system.

The publication related to “The great plate count anomaly (1985)” has been added to revised version as new reference #2.

Weaknesses- My main concern here is the significance of these results outside the model osteocyte system used by this group. Although they carefully avoid over-interpreting their results, there is a strong undercurrent suggesting their approach could enhance aetiologic diagnosis in osteomyelitis and that enumeration of the infecting pathogen might have clinical value. In the first place, molecular diagnostics such as 16S rDNA-directed PCR are well established in identifying pathogens that don't grow. Secondly, it is hard to see how enumeration could have value beyond in vitro and animal model studies since serial samples will rarely be available from clinical cases.

Indeed, we initiated this study for the purpose of trying to improve the diagnostic outcomes for osteomyelitis, in particular that associated with prosthetic joint infection (PJI) but also all other forms, as the current gold-standard diagnostic approaches for this type of infection, either bacterial culture or whole genome sequencing, are very time consuming and costly, and yet are not necessarily accurate. Our method has the benefits (not limited to) of achieving absolute quantification of bacterial load in a shortened time period (in the order of hours) in clinical bone specimens from infected patients. Many of the identified bacterial species in patients were not able to be diagnosed by standard bacterial culturing. Moreover, one of the problematic features of treating bone infection is that repetitive surgeries are usually needed, particularly in PJI, hence, serial clinical bone specimens from the same patient are in fact often available. Therefore, our method of being able to quantify bacterial load offers the advantage of monitoring the infected status throughout the treatment journey. In this study, we chose the *tuf* gene as the targeting sequence to amplify the bacterial signal instead of the well-established *16S* PCR for the reason that *tuf* provides much better sequence discrimination between bacterial species. Therefore, the short PCR amplicon of just 271 bp used in our study, is able to give us a highly accurate taxonomic readout. By this approach, we again shorten the time required for diagnosis. In the last paragraph of the Discussion in the revised manuscript, extra text, a figure demonstrating the strong sequence diversity in *tuf* (Supplementary Figure 2) and an additional reference have been added to address the Reviewer’s concerns.

- I have further concerns regarding the interpretation of the combined bacterial and host cell-directed PCRs against the CFU results. Significance is attached to the relatively sustained genome counts against CFU declines. On the one hand, it must be clearly recognised that the detection of bacterial genomes does not equate to viable bacterial cells with the potential for further replication or production of pathogenic factors. Of equal importance is the potential contribution of extracellular DNA from lysed bacteria and host cells to these results. The authors must clarify what steps, if any, they have taken to eliminate such contributions for both bacteria and host cells. Even the treatment with lysotaphin may have coated their osteocyte cultures with bacterial DNA, contributing downstream to the ddPCR results presented.

We agree that concerns around the interpretation of any molecular readout need to be taken into account. We have yet to find a method that can definitively identify bacterial viability in a clinical setting in the absence of culture. However, PJI and osteomyelitis in general is characterised by a high percentage of culture-negative infection cases, calling for such molecular approaches. Commercially available, so called “live/dead” bacterial PCR reagents exist that act as PCR signal inhibitors by penetrating the cell wall of compromised cells to prevent the PCR signal being generated from those cells. In our experience, while these can provide a certain level of added scrutiny in an experimental setting, they are not definitive because the reaction is often incomplete in an idealised situation and also the reagent may cancel signal from viable bacteria growing under conditions of stress, such as during antimicrobial treatment and host-derived stress imparted in intracellular or intra-tissue environments. Indeed, such stresses are likely contributors to clinical non-culturability. Whole genome sequencing would provide more certainty of bacterial viability to demonstrate genomic intactness but as we discuss herein, this a lengthy and costly process, and one which may prove difficult from host tissue with a low pathogen load. It should be noted that the significance of any diagnostic readout, including from culture, WGS or our method reported here would need to be interpreted by the treating clinical team. We would argue that a rapid, practical molecular diagnostic method in the absence or even presence of culture would provide treating clinicians with an improved rationale for tailoring antimicrobial treatments.

Strengths- On the positive side, the authors provide clear evidence for the value of the direct buffer extraction system they used as well as confirming the utility of ddPCR for quantification. In addition, the successful application of MinION technology to sequence the EF-Tu amplicons from clinical samples is of interest.- Moreover, the phenomenology of the infection studies indicating greater DNA than CFU persistence and differences between the strains and the different MOI inoculations are interesting and well-described, although I have concerns regarding interpretation.